# Quantifying *Plasmodium falciparum* infections clustering within households to inform household-based intervention strategies for malaria control programs: An observational study and meta-analysis from 41 malaria-endemic countries

**Gillian Stresman**[1]*, **Charlie Whittaker**[2], **Hannah C. Slater**[3,4], **Teun Bousema**[5], **Jackie Cook**[6]

**1** Department of Infection Biology, London School of Hygiene & Tropical Medicine, London, United Kingdom, **2** Department of Infectious Disease Epidemiology, London Centre for Neglected Tropical Disease Research and MRC Centre for Outbreak Analysis and Modelling, Imperial College London, London, United Kingdom, **3** MRC Centre for Global Infectious Disease Analysis, Department of Infectious Disease Epidemiology, Imperial College London, London, United Kingdom, **4** PATH, Seattle, Washington, United States of America, **5** Department of Medical Microbiology, Radboud University Medical Center, Nijmegen, The Netherlands, **6** MRC Tropical Epidemiology Group, London School of Hygiene & Tropical Medicine, London, United Kingdom

* Gillian.Stresman@lshtm.ac.uk

## Abstract

### Background

Reactive malaria strategies are predicated on the assumption that individuals infected with malaria are clustered within households or neighbourhoods. Despite the widespread programmatic implementation of reactive strategies, little empirical evidence exists as to whether such strategies are appropriate and, if so, how they should be most effectively implemented.

### Methods and findings

We collated 2 different datasets to assess clustering of malaria infections within households: (i) demographic health survey (DHS) data, integrating household information and patent malaria infection, recent fever, and recent treatment status in children; and (ii) data from cross-sectional and reactive detection studies containing information on the household and malaria infection status (patent and subpatent) of all-aged individuals. Both datasets were used to assess the odds of infections clustering within index households, where index households were defined based on whether they contained infections detectable through one of 3 programmatic strategies: (a) Reactive Case Detection (RACD) classifed by confirmed clinical cases, (b) Mass Screen and Treat (MSAT) classifed by febrile, symptomatic infections, and (c) Mass Test and Treat (MTAT) classifed by infections detectable

**Data Availability Statement:** The data are available as part of this submission in Supporting Information (S1 and S2 Data).

**Funding:** This study was funded as part of a Sir Henry Wellcome fellowship awarded to GS (number 204693/Z/16/Z) from the Wellcome Trust, UK. A Medical Research Council Doctoral Training Studentship (1975152) provided support for CW. The funders had no role in study design, data collection and analysis, decision to publish, or preparation of the manuscript.

**Competing interests:** The authors have declared that no competing interests exist.

**Abbreviations:** CrI, credible interval; DHS, demographic health survey; LAMP, loop mediated isothermal amplification; MCMC, Markov Chain Monte Carlo; MSAT, Mass Screen and Treat; MTAT, Mass Test and Treat; OR, odds ratio; PCR, polymerase chain reaction; RACD, Reactive Case Detection; RDT, rapid diagnostic test; usPCR, ultra-sensitive PCR.

using routine diagnostics. Data included 59,050 infections in 208,140 children under 7 years old (median age = 2 years, minimum = 2, maximum = 7) by microscopy/rapid diagnostic test (RDT) from 57 DHSs conducted between November 2006 and December 2018 from 23 African countries. Data representing 11,349 infections across all ages (median age = 22 years, minimum = 0.5, maximum = 100) detected by molecular tools in 132,590 individuals in 43 studies published between April 2006 and May 2019 in 20 African, American, Asian, and Middle Eastern countries were obtained from the published literature. Extensive clustering was observed—overall, there was a 20.40 greater (95% credible interval [CrI] 0.35–20.45; $P < 0.001$) odds of patent infections (according to the DHS data) and 5.13 greater odds (95% CI 3.85–6.84; $P < 0.001$) of molecularly detected infections (from the published literature) detected within households in which a programmatically detectable infection resides. The strongest degree of clustering identified by polymerase chain reaction (PCR)/ loop mediated isothermal amplification (LAMP) was observed using the MTAT strategy (odds ratio [OR] = 6.79, 95% CI 4.42–10.43) but was not significantly different when compared to MSAT (OR = 5.2, 95% CI 3.22–8.37; $P$-difference = 0.883) and RACD (OR = 4.08, 95% CI 2.55–6.53; $P$-difference = 0.29). Across both datasets, clustering became more prominent when transmission was low. However, limitations to our analysis include not accounting for any malaria control interventions in place, malaria seasonality, or the likely heterogeneity of transmission within study sites. Clustering may thus have been underestimated.

## Conclusions

In areas where malaria transmission is peri-domestic, there are programmatic options for identifying households where residual infections are likely to be found. Combining these detection strategies with presumptively treating residents of index households over a sustained time period could contribute to malaria elimination efforts.

## Author summary

### Why was this study done?

- Malaria is a vector-borne parasitic infection that results in both symptomatic and asymptomatic infections and is a particularly important problem in African, South American, and Southeast Asian countries.

- When malaria transmission becomes low, malaria infections tend to become clustered within populations. In such situations, malaria programs can refine their strategies and begin to target malaria interventions specifically to include household members of malaria-infected individuals detected at the health facility, through community screening of febrile individuals or through mass testing individuals for malaria.

- To make informed decisions on whether, and when, programs should consider household-targeted malaria interventions, evidence is needed on whether malaria infections consistently cluster in households and which strategy is best able to target asymptomatic infections.

## What did the researchers do and find?

- We analysed data from 208,140 African children collected from the DHSs between November 2006 and December 2018 and conducted a meta-analysis of 132,590 individuals of all ages from all malaria-endemic settings around the world published between April 2006 and May 2019.

- Both datasets show that malaria infections do cluster in households at all transmission intensities, but clustering becomes more pronounced as transmission intensity declines (i.e., a larger proportion of infected individuals within a population are clustered in fewer households).

- If all household members of index cases were targeted with interventions, they could potentially treat approximately 75% of all infections in a community once transmission intensity is very low.

## What do these findings mean?

- In locations where local malaria transmission occurs, malaria infections cluster within households of index cases, regardless of how that index case was identified.

- Household targeted strategies, e.g., giving all residents of index households a curative dose of an effective antimalarial drug, provide malaria control programs with an option to easily target infections that otherwise may not be detected.

- In the future, understanding how reactive strategies contribute to achieving malaria elimination will be important in identifying the best strategy and for determining how long it should be sustained.

## Introduction

Malaria transmission is highly heterogeneous between and within populations. Spatial variation in infection risk is evident across the malaria endemicity spectrum but is particularly prominent when transmission is low [1, 2]. Where transmission is peri-domestic (i.e., contact with the mosquito vector occurs around the household considered to be the main residence), data on spatial heterogeneity can assist with planning effective control strategies, through identifying risk factors associated with transmission and targeting of interventions.

Individuals with asymptomatic infections tend to cluster within or around households with a programmatically detectable infection, whether this be a clinical case reporting to a health facility or an infection detected using active screening with routine diagnostic tools such as rapid diagnostic tests (RDTs) [3–6]. The anticipated clustering around such 'index' cases has fuelled the use of reactive strategies as an operational approach to target the asymptomatic parasite reservoir. Reactive strategies become more logistically feasible once transmission is sufficiently low, and this approach has been endorsed by the World Health Organization [4, 6–9]. Despite the biological plausibility, empirical evidence to support this approach is limited: how to best operationalize an effective reactive strategy remains unclear [10].

In low-transmission settings, research and government control programmes have employed various strategies for identifying index households. The most commonly used strategy is Reactive Case Detection (RACD), which involves testing and (if positive) treating household members of each confirmed malaria case passively detected at health facilities [5, 11, 12]. In some settings, testing and treatment activities have been extended to include neighbouring households to account for infectious vectors potentially biting individuals nearby [8, 9]. However, due to the lack of empirical evidence, the radii of screening beyond the index household is typically determined based on available resources, rather than any scientific justification [5, 13]. The primary limitation of the RACD strategy is that it is centred on care-seeking symptomatic cases to flag an index household, which may correspond to a small fraction of all infections, particularly in areas with low treatment-seeking rates or where access to care is limited [14].

Alternative strategies have been employed to address this challenge, including testing anyone in a community with a fever for malaria and treatment of positive cases (Mass Screen and Treat [MSAT]) or testing whole populations for malaria, irrespective of symptoms, to identify and treat infections (Mass Test and Treat [MTAT]) [8]. However, challenges with these strategies as they are currently employed include overlooking asymptomatic infections (MSAT) as well as infections that have parasite densities below the limit of detection of routine diagnostic tools, typically RDTs or microscopy (all approaches) [11, 15]. These low-density infections, which can have a relevant contribution to onward transmission to mosquitoes, are likely to be the most difficult for programmes to identify without relying on more sensitive but impractical molecular diagnostic tools [16]. If low-density infections cluster within the household of detectable infections, it would support the use of household-based reactive strategies involving presumptive treatment of household members and enable targeting of infected individuals who would be missed when relying on routine diagnostic tests. However, there is currently little evidence as to whether household clustering is a consistent feature across settings with different malaria epidemiological conditions (e.g., vector species, population densities, etc.) and whether the extent of this clustering (which would determine the fraction of the undetected reservoir that could be targeted) justifies the use of household-based reactive strategies.

The aim of this work expands the scope of a recent review of RACD to the broader question of whether *Plasmodium falciparum* infections cluster within households of programmatically detectable infections [17]. If infections do consistently cluster within index households, we aim to identify (1) the extent to which residual infections are identified through different infection detection strategies and, in the case of RACD, the utility of extending the targeted radius to neighbours; and (2) at what transmission intensity reactive strategies become most efficacious in terms of the proportion of the parasite reservoir detected. To assess clustering, we evaluated household clustering of malaria infections diagnosed both by routine diagnostic tools (RDT/microscopy) from routinely available demographic health surveillance (DHS) surveys and by molecular methods using a meta-analysis of the available literature.

## Methods

### The trigger for a reactive strategy: Defining index households

The index (case) and control populations were defined according to the 3 main detection strategies currently being implemented: RACD, MSAT, and MTAT (Table 1). Due to the limitations in the DHS data with malaria testing only by RDT/microscopy in children under 7 years of age, related but nonidentical definitions of index and control populations were used across the 2 datasets to assess the degree of infections clustering within households. For the DHS analyses, a child was defined as residing in an index household if the household contained at least 1 other infection detectable through one of the programmatic strategies considered.

**Table 1. Overview of programmatic strategies for identifying households likely to have asymptomatic and/or subpatent infections.**

| Strategy | Index Household | Control Population | Case Definition | Adapted Case Definitions for DHS Data |
|---|---|---|---|---|
| RACD | Residence of individuals with confirmed infections detected within health facilities | 1. Households neighbouring index households with the screening radius defined according to the study<br>2. Households that have been randomly selected in the same community as the index household, as part of a cross-sectional study | All individuals residing in the index household, not including the index case in the case of RACD | Any other child of the household is positive for malaria by RDT/microscopy and has sought antimalarial treatment |
| MSAT | Symptomatic case (fever and RDT/microscopy positive) detected in the community as part of an active campaign | Households that have been randomly selected in the same community as the index household as part of a cross-sectional study | | Any other child of the household positive by RDT/microscopy and has had a fever during the past 2 weeks |
| MTAT | Any infected individual (RDT/microscopy positive) detected in the community as part of an active campaign | | | Any other child of the household is malaria positive by RDT/microscopy |

For this analysis, we have defined an index household, and index and control populations for each strategy. Infections in index and control populations have been confirmed using RDT/microscopy in the DHS studies and molecular methods in the meta-analysis.

**Abbreviations**: DHS, demographic health survey; MSAT, Mass Screen and Treat; MTAT, Mass Test and Treat; RACD, Reactive Case Detection; RDT, rapid diagnostic test

Across the published literature with all-age surveys and molecular tools used for diagnosing malaria, an index household was defined as a household with at least 1 infection detected according to the 3 programmatic strategies. Control populations for the RACD model were defined as either randomly selected households in the community or individuals in neighbouring households (meta-analysis only) depending on the data originally collected. Control households for the DHS data as well as the MSAT and MTAT strategies in the meta-analysis were those in which no individuals had a detectable infection by RDT/microscopy. A review protocol or analysis plan was not prospectively published but is reported in full here, including any adaptations that took place.

## DHS data: Search strategy, selection criteria, and statistical analysis

Individual-level data from 57 DHSs from 23 African countries containing information on malaria infection status (as diagnosed by RDT or microscopy) in children under 7 years of age were collated using the rDHS R package [18]. We extracted data on fever status during the previous 2 weeks (MSAT) and treatment-seeking behaviour (RACD, see S1 Table for detailed definition of treatment-seeking behaviour for each survey) to define index households to mimic the detection strategies being assessed and be consistent with the planned meta-analysis described subsequently (Table 1). Informed parental consent was obained as per the approved DHS study protocols. Using these data, we then defined a series of binary covariates for each child to discern whether they resided in an index household or a control household—specifically, whether an individual shares a household with someone who is malaria positive (MTAT), has recently had a symptomatic malaria infection (MSAT), or has recently sought treatment for malaria (RACD) compared to households negative for each definition. For a detailed description of DHS data management, see S1 Text, and S1 Data for the database.

A logistic regression model was fitted to the child-level data to compare the odds of being malaria positive within an index household compared to those in control households, with the corresponding credible interval (CrI) reported. This model was fitted within a Bayesian

framework, with inference carried out using a Markov Chain Monte Carlo (MCMC)-based sampling scheme in STAN. Uninformative priors were used for all parameters. The model included whether or not the individual lived in an index household (per each of the 3 programmatic strategies), household size, survey prevalence, and an interaction term between survey prevalence and the index household covariate. This allowed the strength of the index household effect to vary with underlying endemicity (see S2 Text for detailed statistical methods).

## Meta-analysis: Search strategy, selection criteria, and statistical analysis

In order to assess the impact of clustering, accounting for subpatent infections, we searched PubMed and included the search terms 'falciparum' and 'epidemiology' and 'polymerase chain reaction' or 'PCR' or 'loop mediated isothermal amplification' or 'LAMP'. Additional searches were conducted using the terms 'malaria' and 'reactive case detection' or 'RACD' as well as 'malaria' and 'mass screen and treat' or 'MSAT' or 'mass test and treat' or 'MTAT'. The search was limited to abstracts written in English or French and published from inception to August 1, 2019, with no limitation to geographic region and with results reported according to the standard guidelines (S3 Text) [19]. Any identified articles that reported primary data on malaria infection in communities were eligible for further screening.

In order to be eligible for inclusion in the meta-analysis, *P. falciparum* malaria infection status had to be determined using routine methods (RDT/microscopy) to define the index status of the household according to one of the reactive models being tested and a molecular method (e.g., polymerase chain reaction [PCR]/loop mediated isothermal amplification [LAMP]) within all household members regardless of age. Informed consent for data collection was obtained as per the original data collection protocol as outlined in the respective publication (S4 Text). The outcome of interest was the odds ratio (OR) of PCR/LAMP-detectable infections (irrespective of RDT/microscopy infection status) identified in index compared to control households, excluding the index case in the RACD strategy. The number of infected individuals in each group was extracted from the published data or requested from the study authors if it was not reported in the format required (see S4 Text for included study references and S2 Data for the database). The studies were grouped according to the programmatic strategy (Table 1). We intended to look at clustering in areas with non-peri-domestic transmission (e.g., forest goers, as defined by the study authors), but only 2 eligible studies—both conducted in Cambodia in populations near forests that explicitly studied forest transmission—were identified. For clarity of message, this model group was removed from the analysis. Only studies in which there was at least 1 infection in each of the index and control populations and for which the setting was expected to have peri-domestic transmission were retained for analysis. Individual-level data were not available for several studies, either on condition of data sharing or because it was extracted directly from the literature, so the meta-analysis focused on the summary estimates per site. The overall OR and—where there were at least 5 studies within a strategy—a strategy-specific pooled OR was calculated using a random effects model, accounting for study, according to the Mantel-Haenszel method. The extent of heterogeneity ($I^2$) between studies was assessed using the DerSimonian-Laird estimator. Analysis was conducted using the metabin function in the meta package (v4.9–6) in R (version 3.6.1) [20]. ORs on the log scale were compared between models using the 't.test' function in R.

## Impact of transmisison intensity and proportion of infections indentified per strategy

To extend the results from the meta-analysis and assess trends of clustering with transmission intensity, we first modelled the linear fit between the log-OR and the inverse of the log-

community-level PCR/LAMP-malaria prevalence, weighting the surveys according to the inverse of the random effect. As it is well acknowledged that routine diagnostics are unable to detect low-density infections, we used the regression model fits from the meta-anlysis to simulate the potential proportion of infections that would be targeted if presumptive treatment was provided to household members living in index households as per programmatic application. We tested community prevalences ranging between 0.01 and 0.5, at 1% increments, and used estimated mean log-transformed ORs and corresponding 95% CIs per detection strategy simulated from 10,000 iterations according to the model fits using a normally distributed random error term, parametized by the mean and standard deviation of the residuals from the log-linear regression model fit. The resulting estimates were back-transformed to the OR scale and summarized according to transmission intensity strata (1%–5%, 5%–15%, and >15%). The OR estimates were then converted to prevalence. To convert the OR to prevalence while ensuring comparability across strategies, we assumed that the prevalence in each control group was 0.2; that the prevalence in index populations was 0.03, 0.1, and 0.25, corresponding to the mean per strata; and a population of 1,000 people.

Ethical approval was not required for this study. The authors had full access to all data in the study and had final responsibility for the decision to submit for publication.

## Results

### DHSs: Evidence for household clustering of patent infections

From 57 DHSs spanning 23 African countries, we extracted data for 72,498, 177,243, and 208,140 (depending on the availability of data to define the index household) children according to the RACD, MSAT, and MTAT strategies, respectively, representing 24,836, 50,590, and 59,050 patent infections (Fig 1, Table 2). Residing in an index household was consistently associated with increased odds of additional infections clustering within the same household. The

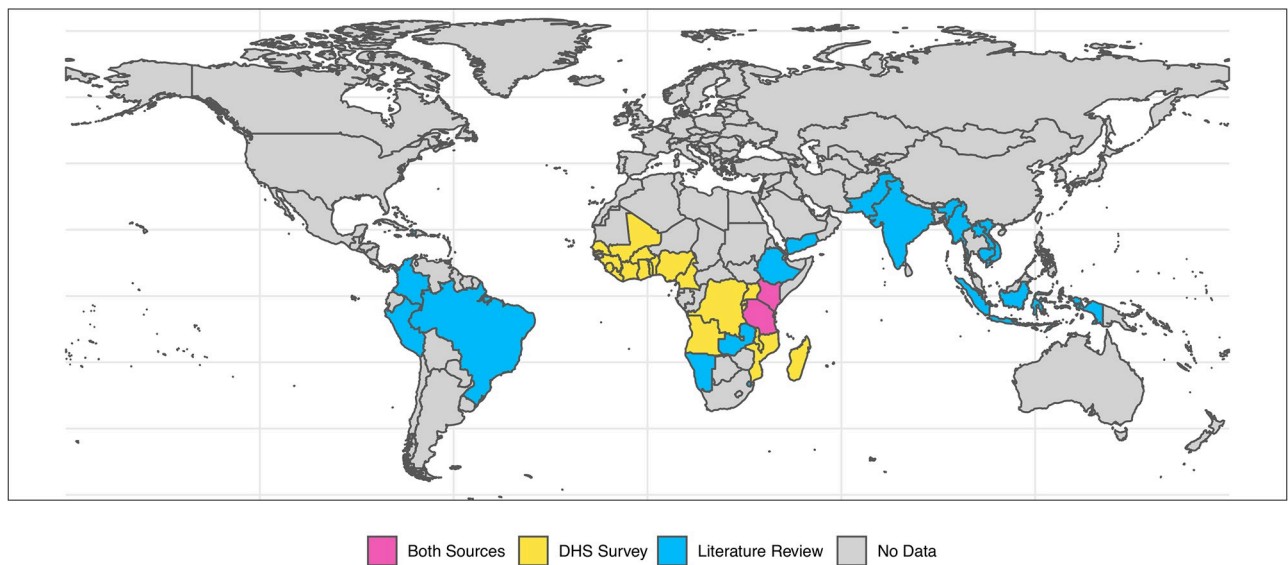

**Fig 1. Global map highlighting the geographic distribution of countries where the data used in the statistical analysis of the DHS data and the meta-analysis were collected.** The map of the world is shown with country boundaries demarcated by black lines. Countries where DHS data were available are shown in yellow (*n* = 23), all in the African region. Countries across the American, African, Middle Eastern, and Asian regrions in blue contributed at least 1 study included in the meta-analysis (*n* = 20). Two countries shown in pink were represented in both analyses. The world map was obtained from the rnaturalearch R package (version 0.1). DHS, demographic health survey.

**Table 2. Summary information providing an overview of the data included in the data analysis using the DHS data and published literature included in the meta-analysis.**

|  | DHS Analysis Data | Meta-Analysis Data |
|---|---|---|
| *N* studies | 57 | 43 |
| **Year data collection (range)** | Nov 2006–Dec 2018 | Apr 2006–May 2019 |
| **Continental region, *N* (*n* unique countries)[a]** | | |
| African | 57 (23) | 17 (7) |
| Americas | 0 | 8 (4) |
| Asia | 0 | 15 (7) |
| Pacific and Middle East | 0 | 3 (2) |
| **All age surveys versus children only, % (*n/N* studies)** | 10.5% (6/57) | 97.7% (42/43) |
| **Median age surveyed (min–max)** | 2 (2–7) | 22 (0.5–100)[a] |
| **Sex, % male (cluster range)** | 50.5 (47.0–52.3) | 54.0 (31.9–100)[b] |
| ***N* individuals sampled (*n* malaria positive)** | 420,659 (59,050) | 132,590 (11,349) |
| **Median age of malaria positive (min–max)** | 3 (2–5) | 22 (0.5–90)[a] |
| **% Male malaria positive (cluster range)** | 50.9 (46.0–53.2) | 47.0 (37.8–56.2)[b] |
| **Number of individuals per model type (*n* malaria positive)** | | |
| RACD versus community | 72,498 (24,836) | 5,661 (288) |
| RACD versus neighbours | 0 | 23,497 (564) |
| MSAT | 177,243 (50,590) | 40,851 (6,238) |
| MTAT | 208,140 (59,050) | 62,581 (4,259) |

[a]22 records did not report age in a relevant format.

[b]10 records did not report gender in a relevant format to extract.

**Abbreviations**: DHS, demographic health survey; MSAT, Mass Screen and Treat; MTAT, Mass Test and Treat; RACD, Reactive Case Detection

greatest degree of this clustering was observed for the MTAT strategy, with an estimated OR of 20.42 (95% CrI 19.36–21.53), although significant clustering was still observed when using the index household definitions for MSAT (OR 14.17, 95% CrI 13.09–15.32) and RACD (OR 11.72, 95% CrI 10.42–13.19). Across all strategies used to define index households, a significant interaction between index household status and overall survey prevalence was observed ($P < 0.001$ in all instances), with clustering of infections becoming more prominent at low transmission levels (Fig 2). Qualitatively similar results were observed using the few DHSs including all age groups as well as a definition of index households identical to that used for the published literature analysis (S1 Text). With the small sample size with all age groups and absence of molecular testing in the DHS data precluding the evaluation of an MTAT-based approach, the results of the proxy definition were the focus here.

## Meta-analysis: Evidence of household clustering of PCR/LAMP infections

The literature search identified 2,754 articles. Of the 279 studies identified for review, 110 met the inclusion criteria. Of the included studies, 10 had published the data in a format in which it could be extracted directly, and 33 studies had either individual or summarized data provided by the authors and were included in the meta-analysis: 18 for RACD, 17 for MTAT, and 8 for MSAT across 35 different study settings representing 11,349 infections in 132,590 individuals (S1 Fig). The included studies were predominantly conducted in African (17) or Asian (15) settings with 8 from South America, 2 in the Pacific, and 1 from the Middle Eastern regions (Fig 1; Table 2). The primary reason for excluding studies was receiving no response to

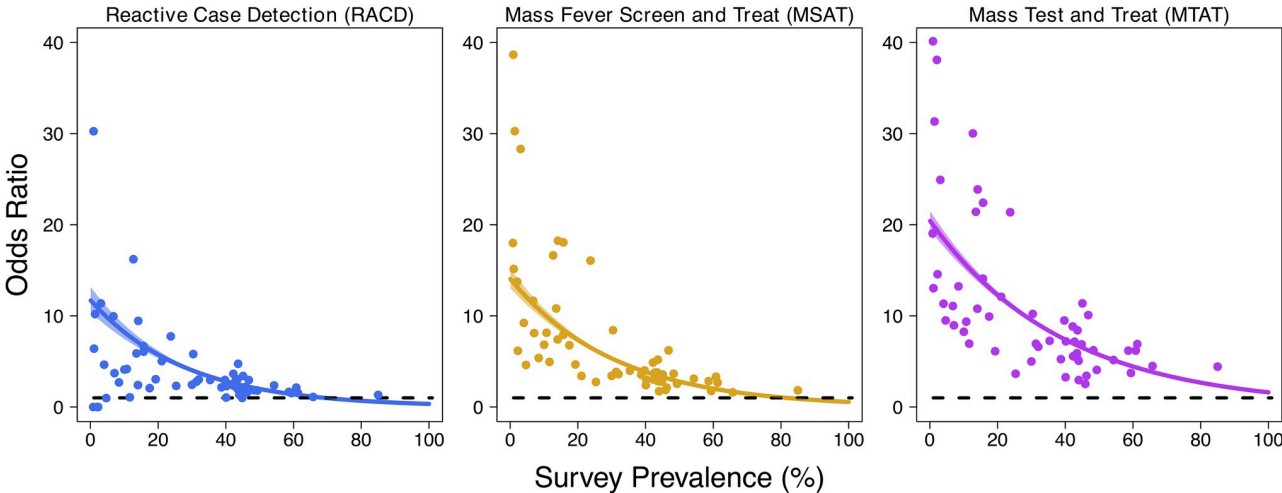

**Fig 2. Malaria infection and the extent of household clustering according to detectability by different programmatic strategies.** DHS data from 57 surveys and 23 countries detailing malaria infection status in children under 7 were collated, and the extent of infection clustering based on detectability by different programmatic strategies (RACD, MSAT or MTAT) was assessed. For each of these analyses, an individual was defined as residing in an index household if the household contained at least 1 other infection detectable through the programmatic strategy considered. The OR of being malaria positive in children who share a household with another child detectable through (A) RACD (clinical care seeking; population-level screening for infecton using routine diagnostics such as RDTs), (B) for the MSAT strategy (population-level recent fever screening), and (C) for the MTAT strategy (population-level screening for infecton using routine diagnostics such as RDTs) as compared to those who do not. Plots display the modelled OR (generated using a logistic-regression approach, coloured line) whilst points are the same ORs but calculated empirically for each survey. Pale shaded area represents the 95% CrI. CrI, credible interval; DHS, demographic health survey; MSAT, Mass Screen and Treat; MTAT, Mass Test and Treat; OR, odds ratio; RACD, Reactive Case Detection; RDT, rapid diagnostic test.

the request for the aggregated data; however, there was minimal evidence of publication bias (S2 Fig).

Across all strategies used to identify an index house (RACD, MSAT, or MTAT), infections detected by PCR/LAMP were more likely to be found within the index household compared to control households (OR = 5.13, 95% CI 3.85–6.84; $P < 0.001$). There were increased odds of detecting additional infections within the household of clinically detected infections (RACD; OR = 4.08, 95% CI 2.55–6.53; $P < 0.001$) (Fig 3). There was also evidence of increased clustering within index households by RACD compared to neighbouring households (OR 2.96, 95% CI 2.06–4.24; $P < 0.001$). There was moderate heterogeneity observed between RACD studies when considering both randomly selected community ($I^2 = 61.7\%$) and neighbour ($I^2 = 59.1\%$) household control populations. A significant linear trend was observed, with the odds of subpatent infections clustering within index households increasing with decreasing transmission with both RACD models (community $P = 0.058$, neighbour $P = 0.009$; Table 3).

The strongest degree of infections clustering in index households was observed using the MSAT (OR = 5.20, 95% CI 3.22–8.37; $P < 0.001$) and MTAT (OR = 6.79, 95% CI 4.42–10.43; $P < 0.001$) strategies (Fig 3). However, the degree of clustering observed according to MTAT was not significantly different from MSAT ($P = 0.883$) or from RACD according to the community ($P = 0.290$) or neighbour ($P = 0.133$) control populations. Clustering of infected individuals within index households increased linearly with decreasing PCR/LAMP prevalence for the MSAT ($P$ for linear trend = 0.001) and MTAT ($P$ for linear trend = 0.097) strategies. There was significant heterogeneity between studies assessed by both MSAT ($I^2 = 96.2\%$) and MTAT ($I^2 = 95.5\%$) detection strategies, likely due to the range of transmission intensity and study designs employed (Table 3).

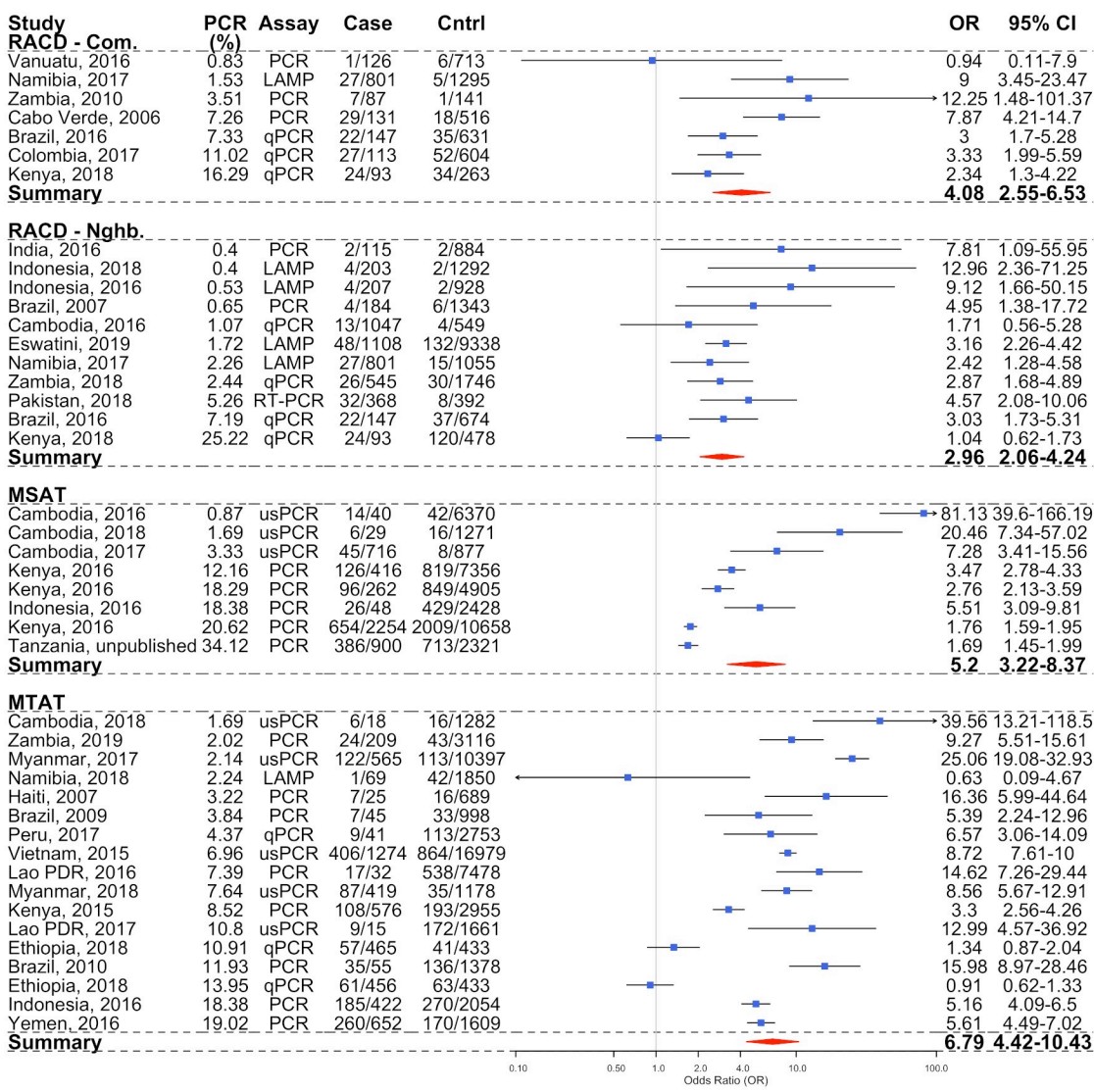

**Fig 3. Forest plot showing the OR of PCR-detectable infections within case compared to control households (blue circles, and OR column), stratified according to the detection strategy.** RACD according to community and neighbour control populations, MSAT, and MTAT with the 95% CIs are shown in the black lines for each OR and in the 95% CI column. The pooled OR statistics for each strategy, as calculated according to a random effects model, with study site as the random effect are shown as red diamonds, with the width representing the 95% CI. Studies are listed in ascending order of malaria PCR prevalence in the community. LAMP, loop mediated isothermal amplification; MSAT, Mass Screen and Treat; MTAT, Mass Test and Treat; OR, odds ratio; PCR, polymerase chain reaction; qPCR, quantitative PCR; RACD, Reactive Case Detection; usPCR, ultra-sensitive PCR.

## Programmatic implications

The overall proportion of all infected individuals in a community that reside within an index house increases as transmission decreases. The OR and corresponding proportion of all infected individuals likely to be residing within the household of an index case were not significantly different across strategies within a given transmission strata (Fig 4; S3 Fig). We estimate that the proportion of infections residing within index households reaches approximately 75% for all strategies when transmission is less than approximately 5% PCR prevalence, but only approximately 30% of all infections when prevalence is above approximately 15% (Fig 4). If a presumptive treatment strategy was applied to all household members within index houses,

**Table 3. Results of the linear regression modelling for each of the 3 strategies assessed, including the neighbor and community controls for the RACD strategy.**

| Strategy | Intercept | Slope, OR (95% CI) | P Value | N Studies | I² (%) |
|---|---|---|---|---|---|
| RACD—Community | 1.04 | 1.78 (1.01–3.12) | 0.058 | 6 | 61.7 |
| RACD—Neighbors | 0.81 | 1.46 (1.13–1.88) | 0.009 | 11 | 59.1 |
| MSAT | 0.80 | 2.34 (1.73–3.17) | 0.001 | 8 | 96.2 |
| MTAT | 1.72 | 1.68 (0.90–3.13) | 0.097 | 17 | 95.5 |

Log linear trends were estimating according to increasing OR against decreasing malaria prevalence. The results are presented on the OR scale for ease of interpretation. PCR method or continental region didn't have an impact on the estimates, but sample sizes were too small to assess this formally. Results suggest an increase in OR or malaria infections clustering as PCR prevalence decreases across all strategies. The number of studies included in the pooled analysis as well as the assessment of heterogeneity within the strata (I²) are also presented.

**Abbreviations**: MSAT, Mass Screen and Treat; MTAT, Mass Test and Treat; OR, odds ratio; RACD, Reactive Case Detection

none of the detection strategies tested was likely to result in targeting of all infected individuals (up to approximately 90% of infected individuals as upper bound estimates), even at the lowest transmission intensity.

## Discussion

To determine whether household-based reactive strategies are likely to result in targeting a substantial proportion of infections in a community, we examined the extent of clustering of malaria infections in relation to programmatically detectable infections. In order to simulate current programmatic practice, we defined index households according to 3 different reactive strategies (RACD, MSAT, and MTAT). The DHS analysis shows that patent malaria infections in children are more likely to be detected within households where another child is infected using all strategies assessed. Furthermore, the results from the meta-analysis indicate that index households are also more likely to have additional infections, including low-density infections that are likely to be undetectable using routine diagnostics. Although the differences in the data collection protocols and analytical methods used precludes a direct comparison, together our results suggest that in the absence of more sensitive field-deployable molecular diagnostic tools, presumptively treating residents of households with a detectable infection could treat a larger proportion of all infections compared to the current test and treat approach employed as part of most reactive activities [13, 21]. Interventions targeting both entire communities or households known to have malaria infections have resulted in short-lived reductions in malaria transmission. [22, 23] However, strategies targeting the household is considered to be operationally sustainable once transmission is low. Given that achieving elimiantion will likely require a sustained effort over the long term, reactive strategies targeting households provides a viable programmatic option.

Given that infections cluster within any index households, the next questions are which of the strategies examined can be recommended and when to switch to targetted strategies. Here, we showed that all strategies were able to identify a similar proportion of all infected individuals in a community when prevalence by PCR is low, when reactive strategies become most practical. However, how best to identify case households will depend on the resources available and the programmatic goals. For example, although dependent on the probability that an infection is accompanied by symptoms that prompt healthcare-seeking behaviour, the RACD model is the least resource intensive with passively detected clinical cases triggering a response [24–26]. If transmission is low and programmatic objectives are to reduce malaria burden, pre-empt a clinical case from developing, or achieve elimination over a longer timeframe, RACD may be the most appropriate strategy. However, if transmission intensity is low to

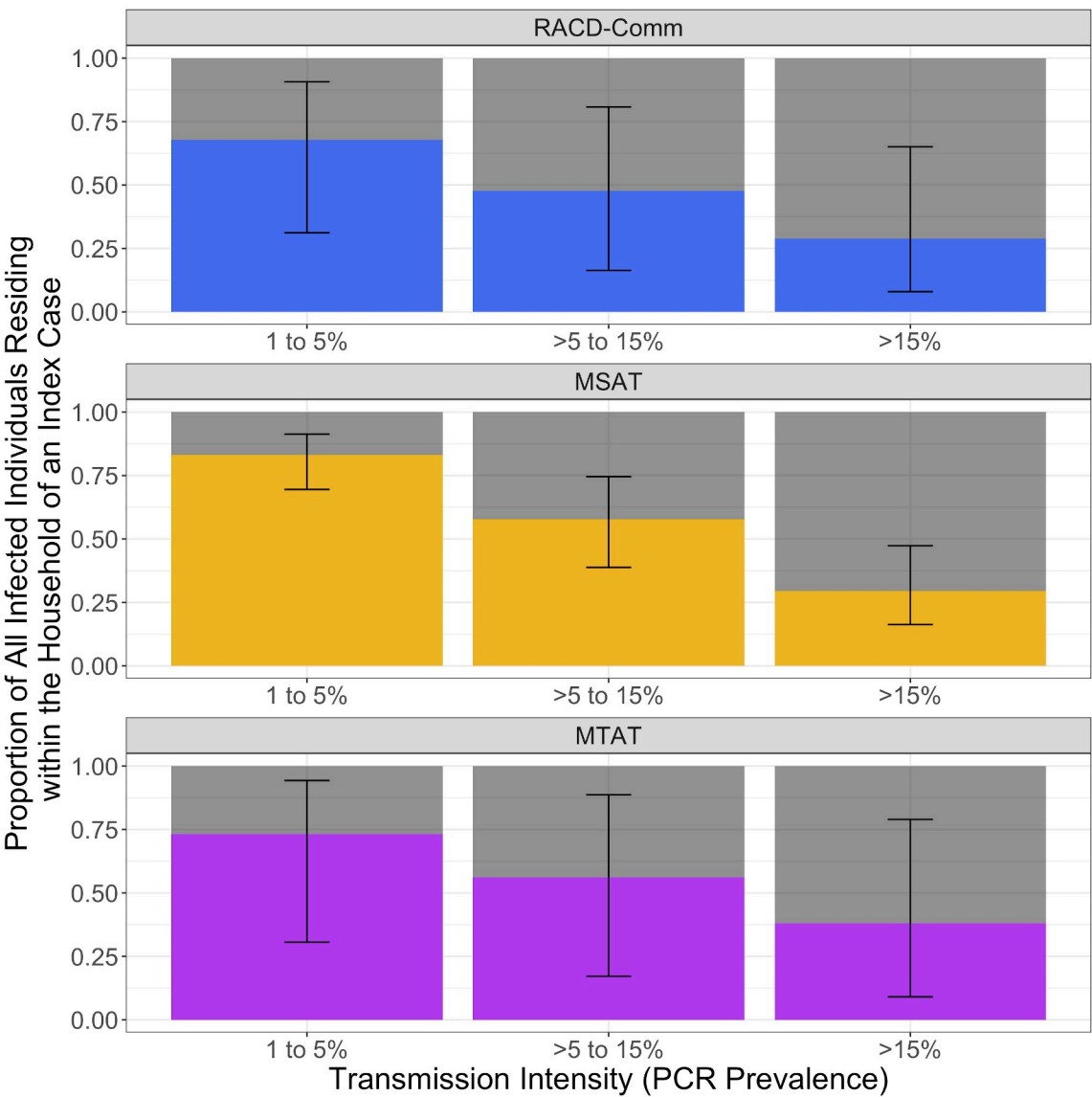

**Fig 4. Bar graphs representing the results from the simulated data showing the proportion of all infections in a community that would likely be detected if RACD, MSAT, or MTAT strategies were employed.** Each panel shows the results for a different reactive strategy with bars ordered by transmission strata according to the PCR prevalence in a community of 1,000 people (e.g., 1% to 5%, >5% to 15%, and >15%). The height of each bar represents the total number of infections within a community, with the colour representing the proportion of infections that would be targeted according to each strategy (black bars corresponding to the 95% CI) with the grey section showing the infections that would be missed. MSAT, Mass Screen and Treat; MTAT, Mass Test and Treat; RACD, Reactive Case Detection.

moderate and the goal is to identify foci within the community or more rapidly reduce transmission while minimizing potentially unnecessary exposure to the intervention (e.g., avoiding community-wide treatment in mass drug administration), the additional resources required to employ the MTAT strategy and presumptively treating members of index households may be justified: Including focal treatment of index households overcomes the main limitation as would target infections for treatment that would be missed by conventional diagnostic tools [27]. Based on the results presented here, if effectively sustained over time, it is plausible that

reactive strategies employing presumptive treatment of index households could contribute to accelerating the road to elimination.

Where neighbours have been included as part of RACD activities, additional infections are typically identified, with some studies showing a significant decrease in positivity with increasing distance [5, 28]. However, the yield of additional infections identified in field studies is low and involves screening a large number of individuals, limiting the operational practicality [29]. Here, we showed that there is some evidence to support infections clustering at the neighbourhood level, which in turn supports the inclusion of neighbouring households as part of the RACD strategy. It is plausible that although few infections were identified in neighbouring households, there still may be more than would be identified in the broader community, supporting their inclusion. However, the individual-level data required to explore this question in more depth—including comparing odds of infection in neighbours to community controls, the optimal screening radii in different settings, and including neighbours as part of MTAT or MSAT strategies (i.e., including neighbours in presumptive treatment)—were not available.

Clustering of infected individuals has been identified in both high- and low-transmission settings, suggesting that household-based strategies to identify residual infections could be appropriate wherever malaria is endemic. The utility and operational feasibility of such an intervention decreases as transmission intensity increases [3, 30]. As transmission increases, reactive strategies become logistically difficult and expensive to implement, making any targeted response options hard to justify [13]. In higher-transmission settings, a smaller fraction of infections are expected to become symptomatic, with even fewer becoming symptomatic enough to prompt care-seeking. As we have highlighted here, RACD, MSAT, or MTAT strategies only identify a small proportion of infected individuals in high-transmission settings, meaning that any targeted intervention would likely result in minimal impact on transmission [24, 29]. Furthermore, when the majority of households are likely to have at least 1 detectable infection such as in higher-transmission areas, uniformly applied interventions (e.g., insecticide-treated bednets) to reduce burden as a first instance are more appropriate [1].

Reactive strategies, such as the ones evaluated in this review, are mainly appropriate where peri-domestic transmission is the main cause of malaria infection. Whilst peri-domestic transmission is common in African settings, in other settings transmission can occur away from the home (e.g., Cambodian forests) [31, 32]. In these settings, clustering within co-travelling populations is prevalent, and identifying appropriate reactive strategies needs to be determined based on the local transmission context within countries (e.g., within Cambodia, both household and forest transmission occurs, depending on the area within the country) [14]. Similarly, in areas with imported infections, any targeted strategies may not be suitable if the area is not receptive to transmission. The same strategies could potentially be initiated to target 'hotpops', but more evidence is needed to to determine which would be the most appropriate strategy [8].

Whilst this study brought together a considerable quantity of data from DHS and published studies with consistent findings identified using both datasets, there are some important limitations to note. The analysis of the DHS data was restricted to infections in children detected by routine diagnostics with strategy definitions modified based on available variables, some of which acted as imperfect proxy indicators, routinely collected as part of the DHSs. However, the results were qualitatively similar to the analysis conducted on the few DHS studies testing all ages, and conclusions were consistent with those of the meta-analysis including all ages and molecular diagnostic tools. Secondly, during the high-transmission season, it is expected that infections are more uniformly distributed across space [1]. Given that the majority of the studies included were conducted during the transmission season (including the DHSs), our resulting ORs may underestimate the extent of clustering. Next, the meta-analysis relied on

aggregated data per study, and therefore we were not able to assess any associated factors that may influence within-household clustering. However, the analysis using the individual-level DHS data did adjust for key factors including household size, and despite the differences in the methods, results were generally consistent, suggesting that there was minimal impact on the overall conclusions. The differences between the DHS and published literature aggregated to conduct the meta-analysis in terms of target population, data available, geographic spread, and analytical methods employed were such that a direct comparison between the resulting ORs is not meaningful. However, given the trends of increasing clustering as transmission decreased were qualitatively similar, the analysis is nevertheless informative. Next, our definition of peri-domestic transmission was based on the epidemiological conditions as reported by the authors of the original publication. It is possible that included studies were misclassified as being peri-domestic transmission. However, impact of misclassification would be expected to attenuate the pooled OR and therefore is expected to have minimal impact on the interpretation of the results. Finally, all neighbours were grouped together regardless of distance from the RACD index household despite there being a known distance decay function in terms of the degree of clustering [28]. This may have diluted the magnitude of clustering observed.

Ultimately, malaria infections cluster within households where programmatically detectable infected individuals reside, with the magnitude of clustering increasing as transmission intensity decreases. At low transmission, if effectively implemented, all strategies assessed are expected to detect households where the majority of infected individuals are likely to reside. Programmatic options are available that can act as signposts to target focal treatment interventions within households of detectable infections, providing an alternative option to mass treatment campaings (e.g., when risk of drug side effects outweights the risk of malaria infection). Such activities have the potential to enhance elimination efforts and, in combination with effective vector control and strong health systems, may contribute to reducing transmission if a sufficient proportion of all infections can be targeted.

## Supporting information

**S1 Table. Definition and extraction of treatment-seeking variables.**
(DOCX)

**S2 Table. Results of the logistic regression modelling for each of the 3 models assessed using the DHS datasets.**
(DOCX)

**S1 Text. Detailed methods for DHS data extraction.**
(DOCX)

**S2 Text. Statistical analysis and regression modelling of DHS data.**
(DOCX)

**S3 Text. PRISMA checklist.** PRISMA, Preferred Reporting Items for Systamtic Reviews and Meta-analyses.
(DOCX)

**S4 Text. Reference list for published studies included in the meta-analysis.**
(DOCX)

**S1 Fig. Flow diagram providing and overview of the results from the literature search to identify eligible studies included in the meta-analysis.**
(DOCX)

**S2 Fig. Funnel plot for visual inspection of publication bias of studies included in the meta-analysis.**
(DOCX)

**S3 Fig. The proportion of all infections in a community that would likely be detected if RACD, MSAT, or MTAT strategies were employed by transmission strata.** Each panel shows the results of the simulated data for different transmission strata according to the PCR prevalence in a community of 1,000 people (e.g., 1% to 5%, >5% to 15%, and >15%). The height of each bar represents the total proportion of infections within a community, with the colour representing the proportion of infections that would be detected according to each strategy (and corresponding uncertainty) with the grey section showing the infections that would be missed.
(DOCX)

**S1 Data. Steps taken and variables used in the management of the DHS datasets used in the analysis.**
(XLSX)

**S2 Data. Aggregate, study-level data collated from the studies included in the meta-anlaysis.**
(CSV)

## Author Contributions

**Conceptualization:** Gillian Stresman, Charlie Whittaker, Hannah C. Slater, Teun Bousema, Jackie Cook.

**Data curation:** Gillian Stresman, Charlie Whittaker.

**Formal analysis:** Gillian Stresman, Charlie Whittaker.

**Investigation:** Gillian Stresman, Charlie Whittaker.

**Methodology:** Gillian Stresman, Charlie Whittaker, Hannah C. Slater, Jackie Cook.

**Project administration:** Gillian Stresman.

**Resources:** Gillian Stresman.

**Supervision:** Hannah C. Slater, Jackie Cook.

**Writing – original draft:** Gillian Stresman.

**Writing – review & editing:** Gillian Stresman, Charlie Whittaker, Hannah C. Slater, Teun Bousema, Jackie Cook.

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
