## [Editor Report · Decision Letter 0]

21 Feb 2020

Dear Dr Stresman, 

Thank you for submitting your manuscript entitled "Do Plasmodium falciparum infections cluster at the household level? A meta-analysis of the evidence to inform household based intervention strategies for malaria control programs" for consideration by PLOS Medicine.

Your manuscript has now been evaluated by the PLOS Medicine editorial staff [as well as by an academic editor with relevant expertise] and I am writing to let you know that we would like to send your submission out for external peer review.

Please re-submit your manuscript within two working days, i.e. by 25th Feb 2020, 11:59PM.

Kind regards,

Clare Stone, PhD

PLOS Medicine

---

## [Decision Letter · Decision Letter 1]

26 Jun 2020

Dear Dr. Stresman,

Thank you very much for submitting your manuscript "Do Plasmodium falciparum infections cluster at the household level? A meta-analysis of the evidence to inform household based intervention strategies for malaria control programs" (PMEDICINE-D-20-00366R1) for consideration at PLOS Medicine. 

[LINK]

In light of these reviews, I am afraid that we will not be able to accept the manuscript for publication in the journal in its current form, but we would like to consider a revised version that addresses the reviewers' and editors' comments. Obviously we cannot make any decision about publication until we have seen the revised manuscript and your response, and we plan to seek re-review by one or more of the reviewers. 

We expect to receive your revised manuscript by Jul 17 2020 11:59PM. Please email us (plosmedicine@plos.org) if you have any questions or concerns.

We look forward to receiving your revised manuscript. 

Sincerely,

Clare Stone, PhD

Managing Editor 

PLOS Medicine

plosmedicine.org

Please revise your title according to PLOS Medicine's style. Your title must be nondeclarative and not a question. It should begin with main concept if possible. "Effect of" should be used only if causality can be inferred, i.e., for an RCT. Please place the study design ("A randomized controlled trial," "A retrospective study," "A modelling study," etc.) in the subtitle (ie, after a colon). Please also add a country setting.

In the abstract – please indicate dates around the data sets and cohort analysis and also some geographical information. We in addition need summary demographic information. “; Please add p values along with the provided 95% Cis here in the abstract and throughout including tables. The final sentence of the Methods and Findings section of the abstract should include a sentence or two on the limitations of the study. 

Did your study have a prospective protocol or analysis plan? Please state this (either way) early in the Methods section. a) If a prospective analysis plan (from your funding proposal, IRB or other ethics committee submission, study protocol, or other planning document written before analyzing the data) was used in designing the study, please include the relevant prospectively written document with your revised manuscript as a Supporting Information file to be published alongside your study, and cite it in the Methods section. A legend for this file should be included at the end of your manuscript. b) If no such document exists, please make sure that the Methods section transparently describes when analyses were planned, and when/why any data-driven changes to analyses took place. c) In either case, changes in the analysis-- including those made in response to peer review comments-- should be identified as such in the Methods section of the paper, with rationale.

It seems this analysis is using data from Children, this needs to be added to the title, if so, the abstract and made explicitly clear in the main text – it’s not until Table 1 that this was apparent to me. The reader needs to understand this from the start. It is slightly confusing atm as you then say the malaria status of all household members are tested. If this is not the main part of the analysis, please be explicit. 

There is a lack of relevance in the main text about the locations and geographic regions that are being surveyed. Please add. 

Was written consent provided? 

Please provide a table with demographic information (main text)

Please ensure that the study is reported according to the STROBE guideline, and include the completed STROBE checklist as Supporting Information. 1 Please add the following statement, or similar, to the Methods: "This study is reported as per the Strengthening the Reporting of Observational Studies in Epidemiology (STROBE) guideline (SChecklist)." The STROBE guideline can be found here: http://www.equator-network.org/reporting-guidelines/strobe/ When completing the checklist, please use section and paragraph numbers, rather than page numbers.

Comments from the reviewers:

Reviewer #1: The manuscript by Stresman et al. "Do Plasmodium falciparum infections cluster at the household level? A meta-analysis of the evidence to inform household based intervention strategies for malaria control programs"

Major

I full-heartedly agree with authors' claim that "Reactive malaria strategies are predicated on the assumption that individuals infected with malaria are clustered within households or neighbourhoods. Despite the widespread programmatic implementation of such strategies, little empirical evidence exists as to whether such strategies are appropriate and if so, how they should be most effectively implemented." Unfortunately, authors fail to provide information that were not found in the previous studies or give different angle to the issue. 

I do not think there is a single intervention that works on its own, and household- based intervention strategies are not exception. It is yet unclear to me at what level, or stage of transmission intensity, household-based intervention can actually contribute to elimination. Different approaches such as mass drug administration (The Gambia. Mwesigwa et al., 2019), or indoor residual spraying (Uganda. Nankabirwa et al., 2019) that are more intensive seem to result in short lasting effect. 

Line 102-4 "However, there is currently little evidence as to whether household clustering is a consistent feature of malaria micro-epidemiology and whether the extent of this clustering (which would determine the fraction of the undetected reservoir that could be targeted) justifies the use of household-based reactive strategies." - I thought clustering at different socio-spatial level, such as household, is indeed a part of malaria micro-epidemiology…? Readers may benefit from some clarification with what you are referring with malaria micro-epidemiology in this sentence. 

Line 362 "if a sufficient proportion of all infections can be targeted." - I think this is the fundamental question we are all interested in, but I did not see this aspect discussed in discussion. 

Minor

Using antimalarials use as a proxy for seeking care at health facilities - In order for this assumption to work well, it is important that antimalarials are available at health facilities and not at private pharmacies. Among 23 countries DHS surveys were obtained from, do you have information on how antimalarials are distributed? 

Figure 2 - Please use image with higher resolution

Line 326: "were" should be "where"? 

Reviewer #2: Dear Authors,

In this paper the authors investigated to what extent Plasmodium falciparum infections cluster at the household level based on existing literature data for both patent and sub-patent infections. 

The review was conducted in a comprehensive way and collates evidence which indicates to what extent the crucial, but not yet rigorously tested, assumption of clustering of malaria positive individuals at the household level, holds. National Malaria programs in different countries include spatially targeted interventions based on the assumption that indeed some form of spatial clustering at the household (or neighboring households) is present. The data demonstrated extensive clustering for both patent and sub-patent infections at the household level and increasing odds of finding positive cases at index households with decreasing prevalence. Moreover they point out that in areas where malaria is peri-domestic the existing programmatic options are sufficient to identify those households where residuals infections are likely to be found. They suggest that once identified treating all members of such index or 'source' households could provide an alternative to mass treatment campaigns and potentially enhance elimination efforts. 

In summary I think the authors did an excellent job and the paper certainly fits the criteria for publication in PloS Medicine. 

However I would like to point out two major aspects which could warrant some more discussion. 

Major comments:

Peri-domestic transmission is somewhat loosely defined as malaria transmission which occurs around the household. The authors point out in their discussion that peri-domestic transmission is common in the African setting, but not always so in other places. For example in a Cambodian forest setting transmission can occur far away from home (typically associated with a dual housing setting). The authors did not emphasis in the main text that the entire DHS survey is compiled of countries from the African continent, whereas the PCR/LAMP meta-analysis is a mix of countries from different regions. They did state however on line 177 that they retained those studies which were expected to have peri-domestic transmission, and also in table 2 that 'region' didn't affect the results. However, it is unclear how these choices were made, what exactly was their basis for deciding that transmission is expected to be peri-domestic (or not)? Or perhaps more informative which studies were not retained on the basis of expecting non peri-domestic transmission. This info could be included in the supp material, e.g. the flowchart of the meta-analysis, extended figure 3. It would be of interest from a more programmatic point of view to see whether or not these omitted studies are all from one country, one continent, … as well as some additional info on what they do consider as peri-domestic and what not. For example in Cambodia, farmers often have one residence in the village and an additional small house several km away at their field plots, in which they also spend the nights, especially when crop planting/harvesting needs to be done. This system is much more outspoken in the north-east of Cambodia versus the western provinces (Pailin, Battambang), but to my knowledge, not absent. So on what basis could a program manager decide that his/her setting is sufficiently 'peri-domestic' to initiate a spatial intervention based on index households determined from programmatic info available? 

The second point I want to raise is regarding the differences in odds ratio of the patent versus sub- patent infections. As the DHS survey is compiled of African countries, generally having higher prevalence rates, while the sub-patent data are comprised of countries on three continents having generally lower prevalence, what could have driven a two- to nearly three-fold increase in odds ratio ('clustering') in the DHS data versus the PCR/LAMP data? Does this simply reflects the way the analysis is performed or could there be other reasons? Perhaps the authors could comment on this apparent contradiction in the text as well. 

Minor comments.

L39: typo: detection instead of detecation

L136: It is striking that all data collected from 23 DHS surveys are all countries from the African continent, therefor add '23 African countries' here.

L177: On what basis exactly was a study designated as being non peri-domestic, and how many such studies were excluded?

L151: Define the uninformative priors in the S2 text exactly, what distribution was used?

L156 & S2 text: Did you truly fit a no-intercept regression? If not add 'beta 0' as intercept coefficient. In fact I would suggest you simply extract the STAN code, and add it in S2 text, it will become clear immediately how the model was fitted (as well as what priors were used). 

L157/158: Upon first read it is simply not clear why you use an individual-level approach for the DHS data, but need a meta-analysis for the sub-patent infections of the PCR/LAMP data. I suggest to clarify this briefly in one/two sentences just before L158.

L169: should be RDT or PCR, instead of AND?

L197: prevalence instead of prevalanece

L207: add 'African' countries

L213: CI is here a credible interval, not a confidence interval given you did use MCMC. However on the same page L192 you mention confidence interval. You need to distinguish between the two as they, from a statistical point of view, are not the same. 

L244 - table 2: With region it seems the author indicate 'continent', or 'continental region', please adapt.

L357: individuals misspelled.

L416: type: fine not find-scale

Reviewer #3: I confine my remarks to statistical aspects of this paper. These were well done and I recommend publication.

Peter Flom

[LINK]

---

## [Decision Letter · Decision Letter 2]

12 Aug 2020

Dear Dr. Stresman,

Thank you very much for re-submitting your manuscript "Quantifying the clustering of Plasmodium falciparum infections within households to inform household based intervention strategies for malaria control programs: A statistical and meta-analysis from 41 malaria endemic countries" (PMEDICINE-D-20-00366R2) for review by PLOS Medicine.

I have discussed the paper with my colleagues and the academic editor and it was also seen again by reviewers. I am pleased to say that provided the remaining editorial and production issues are dealt with we are planning to accept the paper for publication in the journal.

[LINK]

We look forward to receiving the revised manuscript by Aug 19 2020 11:59PM. 

Sincerely,

Clare Stone, PhD

Acting Chief Editor 

PLOS Medicine

plosmedicine.org

Requests from Editors:

Title: "statistical and meta-analysis ..." looks a bit too word, but also quite unusual, I suggest removing "statistical and" (or perhaps more appropriate, altering to "...: an observational study and meta-analysis ...")

Abstract - the limitations could be better signposted in the abstract, please.

Line 42 – in which countries – please state early in the abstract what the geographical setting is, also we do need the demographic information, as previously discussed. 

Line 43 and elsewhere, please add in exact dates rather than just years.

Main text, please tell us the countries the surveys are conducted in – generally the manuscript feels ‘lite’ on information around the geographical and demographic information throughout. Please add to avoid further rounds of review.

Methods – please add a note in the methods section that ethics approval was not required (if so)

remove "role of the funding source" at line 260

Lines 453-462 – remove as this information will be pulled from the meta data in the submission form

Comments from Reviewers:

Reviewer #1: Dear authors, 

Thank you for addressing questions and suggested edits. I do not have any further recommendations at this point and I think the paper is ready to be published. 

Reviewer #2: The authors have now appropriately addressed the various remarks from the reviewers and I recommend the paper for publication. 

I did note one last typo on line 71: 'situtaions', please correct.

[LINK]

---

## [Editor Report · Decision Letter 3]

11 Sep 2020

Dear Dr. Stresman, 

On behalf of my colleagues and the academic editor, Dr. Lorenz von Seidlein, I am delighted to inform you that your manuscript entitled "Quantifying Plasmodium falciparum infections clustering within households to inform household-based intervention strategies for malaria control programs: An observational study and meta-analysis from 41 malaria endemic countries" (PMEDICINE-D-20-00366R3) has been accepted for publication in PLOS Medicine. 

PRODUCTION PROCESS

PRESS

PROFILE INFORMATION

Thank you again for submitting the manuscript to PLOS Medicine. We look forward to publishing it. 

Best wishes, 

Clare Stone, PhD

Managing Editor 

PLOS Medicine

plosmedicine.org